# Arabica coffee Intercropped with *Urochloa decumbens* Improved Nutrient Uptake and Yield in the Brazilian Cerrado

**DOI:** 10.3390/plants14040496

**Published:** 2025-02-07

**Authors:** Thais Rodrigues de Sousa, Arminda Moreira de Carvalho, Maria Lucrecia Gerosa Ramos, Douglas Rodrigues de Jesus, Ana Caroline Pereira da Fonseca, Fernanda Rodrigues da Costa Silva, Alexsandra Duarte de Oliveira, Heloisa Carvalho Ribeiro, Adriano Delly Veiga, Robélio Leandro Marchão, Raíssa de Araujo Dantas, Fabiana Piontekowski Ribeiro

**Affiliations:** 1Faculty of Agronomy and Veterinary Medicine, University of Brasília, Campus University Darcy Ribeiro, University of Brasília-UnB, Brasília 70910-970, Distrito Federal, Brazil; lucrecia@unb.br (M.L.G.R.); 231106065@aluno.unb.br (D.R.d.J.); 231105578@aluno.unb.br (A.C.P.d.F.); 231106074@aluno.unb.br (F.R.d.C.S.); 2Embrapa Cerrados, BR 020, Km 18, Planaltina 73310-970, Distrito Federal, Brazil; alexsandra.duarte@embrapa.br (A.D.d.O.); adriano.veiga@embrapa.br (A.D.V.); robelio.marchao@embrapa.br (R.L.M.); raissa.dantas@colaborador.embrapa.br (R.d.A.D.); fabiana.ribeiro@unb.br (F.P.R.); 3Department of Forestry Engineering, Campus Darcy Ribeiro, University of Brasília, Brasília 70910-970, Distrito Federal, Brazil

**Keywords:** intercropping system, nutritional requirement, mineral nutrition, cover crop

## Abstract

Coffee intercropped with *Urochloa decumbens* modifies nutrient uptake, and consequently the yield and quality of coffee, by the greater release of nutrients and efficient nutrient cycling. There is little information about the increasing nutrient content in Arabica coffee plants intercropped with *Urochloa decumbens*. The objective of this study was to evaluate the effect of *Urochloa decumbens* intercropped with two coffee cultivars (*Coffea arabica* L.) on the levels of macro- and micronutrients and coffee crop yield. The experiment was conducted at Embrapa Cerrados, Planaltina-DF, and was arranged in a completely randomized block design with three replications, in a factorial design. The first factor consists of two management systems: with (WB) and without (NB) *Urochloa decumbens* intercropped; the second factor is composed of Arabica coffee cultivars ‘IPR-103’ and ‘IPR-99’. There was no significant difference in yield of the coffee cultivars with and without *Urochloa decumbens* intercropped between the rows. The treatment with ‘IPR-99’ coffee cultivar intercropped with *Urochloa decumbens* achieved 400 kg ha^−1^ (8 bags) more than the other treatments. The presence of *Urochloa decumbens* increased leaf nutrient contents of the macronutrients Ca and Mg and micronutrients Mn and Fe. Thus, the Arabica coffee–*Urochloa decumbens* intercropping system is an efficiency management strategy which improves nutrient content for the main crop with consequent yield gains.

## 1. Introduction

Coffee is one of the key commodities in the world, mainly occurring in tropical regions, and contributes markedly to the income of small-scale farmers [1]. Of the 130 coffee species, *Coffea arabica* L. is the most commonly cultivated [2] and most widely consumed (50% of all coffee consumed) in the world [3]. In Brazil, the crop is socio-economically important and still has a great potential for expansion. In addition to the direct jobs, the sector generates indirect employment and plays an important role in the Gross Domestic Product (GDP) of Brazil, due to the high coffee output and export quality [4]. Between January and August 2024, Brazil exported 32.1 million bags, i.e., an increase of 40.1% compared to the same period in 2023 [5].

Although the Brazilian Cerrado has a low natural soil fertility, coffee production was made possible in that region [6] by combining the favorable edaphoclimatic conditions with irrigation and soil fertilization [7,8,9]. However, to minimize negative effects that restrict the root development and nutrient uptake of the coffee trees, soil management practices are crucial [9]. The genetic variability of the coffee species influences the nutritional status, nutrient accumulation, use efficiency, and content in leaves and branches, in addition to water availability [10,11,12], which is closely related to nutrient uptake.

Intercropping systems represent a management strategy by which one can improve aggregate stability and, consequently, soil structure, which affects total organic carbon accumulation and carbon stock [8,13], which may increase crop yields by improving nutrient uptake and reducing environmental impacts. One of the challenges of modern agriculture is to provide higher crop yields with minimal environmental impact, with a view to climate change mitigation [14,15].

In the Brazilian coffee industry, producers are greatly concerned about socio-environmental impacts and about being able to warrant sustainable coffee production [16]. The implementation of new management strategies, e.g., intercropping systems, has contributed to nutrient cycling by the cultivation of cover crop species in the interrow, avoiding losses of nutrients in the system, and increasing soil quality with higher soil organic matter contents, which directly affects nutrient availability for plants. In this way, the sector has not only achieved higher yields and profitability but also enhanced environmental conservation [8,13,17].

Leaf analysis is one of the main tools by which the nutritional status of crops can be assessed to understand nutrient cycling in the system. In this way, nutrient deficiency or excess in plants can be detected, and soil analysis can contribute to management strategies for plant nutrition [18,19,20]. Nitrogen is the nutrient required most for coffee and fertilization recommendations can exceed 550 kg N ha^−1^ [21].

In intercropping systems of coffee with *Urochloa decumbens*, the Poaceae shoots are cut off periodically at a plant height of 60 cm [8,13]. The estimated dry biomass of 10 t ha^−1^ of *Urochloa decumbens* benefits coffee crops. The decomposition process influences biotic and abiotic factors and, consequently, nutrient cycling and crop yield [22,23]. These factors influence nutrient uptake by coffee in intercropping systems [24].

The authors of this study hypothesized that in the Brazilian Cerrado, the inclusion of *Urochloa decumbens* intercropped with Arabica coffee would increase leaf nutrient contents and coffee yield and, consequently, the quality of Arabica coffee by a greater release of nutrients and efficient nutrient cycling. To this end, we evaluated the leaf nutrient content and yield of two Arabica coffee cultivars (cvs. ‘IPR-103’ and ‘IPR-99’), intercropped or not with *Urochloa decumbens*, under irrigation, in the Brazilian Cerrado.

## 2. Results and Discussion

### 2.1. Chemical Composition of Urochloa Decumbens and Arabica coffee Yield

There was no significant difference in plant dry weight production between the 10 cuts of *Urochloa decumbens* intercropped with coffee cultivars, resulting in a mean dry matter accumulation in the first cut of 3.34 t ha^−1^ for ‘IPR-103’ and 4.26 t ha^−1^ for ‘IPR-99’, and values in the last cut of 3.30 t ha^−1^ for ‘IPR-103’ and 2.63 t ha^−1^ for ‘IPR-99’ (Figure 1).

The structural composition and nutrient contents (macro and micronutrients) in the shoots of *Urochloa decumbens* are listed in Table 1. The mean contents of cellulose and hemicellulose were 263.9 and 314.8 g kg^−1^, respectively, and the lignin and lignin/N contents were 30.6 and 1.6 g kg^−1^, respectively. The results were considered favorable for a more accelerated decomposition of plant residues [22,25], promoting faster nutrient cycling and thus increasing the soil C pool [25].

The chemical composition of plant tissue of *Urochloa decumbens* compared with the soil organic matter (SOM) pool confirmed the hypothesis that, after eight years of cultivation in the Cerrado biome, cover crops with a higher shoot concentration of soluble fractions, e.g., hemicelluloses, and with a lower lignin concentration and lignin/N ratio, promoted a marked accumulation of SOM labile fractions and soluble humic substances, such as fulvic and humic acids, and, consequently, greater soil C stocks [25].

Regarding the macronutrients in *Urochloa decumbens*, the highest levels were found for K, followed by N, and the lowest content was found for S. For the micronutrients, the concentration of Fe was highest and Cu was the lowest (Table 1). The macronutrient levels were within the standard considered adequate for *Urochloa decumbens* or even above the range, the values for which were as follows: 12–20 g kg^−1^ for N, 2–6 g kg^−1^ for Ca, 1.5–4 g kg^−1^ for Mg, 0.8–3 g kg^−1^ for P, and 12–25 g kg^−1^ for K, according to [26]. The highest K levels in *Urochloa decumbens* can be explained by the high capacity of the plants to concentrate and accumulate this macronutrient, being able to promote K cycling in the system by the vigorous root system of the species of this genus [27,28]. N content in crop residues of *Urochoa decumbens* may contribute to N uptake by plants if the residue contains approximately 17 g N kg^−1^ [29].

With regard to micronutrients, it is necessary to analyze whether the inputs efficiently achieve the critical levels which would influence crop yields. The concentrations of the micronutrients B, Fe, and Mn were considered above the adequate range for *Urochloa decumbens.* The adequate ranges are 10–25 mg kg^−1^ for B, 50–250 mg kg^−1^ for Fe, and 40–250 mg kg^−1^ for Mn, according to reference values [26]. The high concentration of these micronutrients in the leaf of *Urochloa decumbens* shows the efficiency of the species in increasing soil micronutrients’ availability through the release of siderophores, avoiding nutrient uptake [27]. This assessment is essential to detect if the inputs are possibly not efficient in attaining the critical nutrient levels, both in terms of plant production and nutrient competitiveness in intercropping systems.

Despite a positive difference of 8 bags ha^−1^ in cultivar ‘IPR-99’ with *Urochloa decumbens* between the rows, there was no significant difference in relation to the yield of the Arabica coffee cultivars in the treatments with and without *Urochloa decumbens* intercropped between the rows (Figure 2). The yield of these cultivars (cvs ‘IPR-103’ and ‘IPR-99’) exceeded 80 bags ha^−1^ (bags of 60 kg), in both treatments, reaching approximately 95 bags ha^−1^ in cv. ‘IPR-99’, in the intercropping treatment with *Urochloa decumbens*.

Cultivars ‘IPR-103’ and ‘IPR-99’ are considered to have high yields and good adaptability to the conditions of the Cerrado biome, with excellent vegetative vigor. In studies in the Brazilian Cerrado, cultivar ‘IPR-103’ yielded over 65 bags ha^−1^, which demonstrates the high adaptability to Cerrado conditions, similarly to cultivar ‘IPR-99’, with excellent grain yield [30]. Therefore, in this study, the yields of both ‘IPR-103’ and ‘IPR-99’ were higher than reported in the literature [31].

Grain yield is related, among other factors, to soil quality, which is the result of the soil physical, chemical, and biological properties [32]. Therefore, although in this study no significant difference was observed in the intercropping management of Arabica coffee and *Urochloa decumbens*, the use of cover crops such as *Urochloa decumbens* in coffee production systems may increase the production capacity, owing to the improved soil quality [33], resulting from higher SOM contents and enzymatic activity [8,34,35,36,37].

### 2.2. Leaf Nutrient Content

The function that best describes the variable nutrient concentrations (of N, Ca, K, P, Mg, S, Cu, Mn, Fe, and B) is the polynomial model, for both studied treatments (with and without *Urochloa decumbens* between the rows). For N contents, the model coefficients were significant for the systems evaluated at 5% significance, with the exception of ‘IPR-103’ NB. For P and Ca contents, the model coefficients were significant for all systems evaluated at 5% significance, while for K, Mg, and S, the coefficients were not significant at 5% significance (Appendix A).

Most of the evaluated macro- and micronutrients were influenced by seasonal variation, which may be related to the development stages of the Arabica coffee cultivars, as mentioned by Oliosi et al. [12].

The coffee N contents varied over time, from 13 to 29 g kg^−1^ at the beginning of the evaluation (water-stressed environment); and from 25 to 32 g kg^−1^, after 60 days, under full irrigation and fertilization, when the plant grows new leaves, and under stable conditions, where it will actively develop leaves and form leaf buds. At 210 days, there was a decline in N levels 25–32 g kg^−1^. The experiment began at the end of the controlled water stress period and the N levels were considered low at the beginning of the evaluation of N levels in cv. ‘IPR-103’ NB 8.13 g kg^−1^. After 60 days (period corresponding from November 2021 to January 2022), the ideal range, called the “adequate range”, is achieved 25 to 30 g kg^−1^, as recommended by Cantarella [38] (Figure 3).

The N content was higher for cultivar ‘IPR-103’ NB at 180 days (after fruit filling) than ‘IPR-103’ WB and ‘IPR-99’ WB (*p* > 0.05). These levels were similar to the maximum levels established by Ferreira et al. [39]. The increase in leaf N content between the evaluation dates could be explained by increasing rates of nitrogen fertilizers, but this was not the case in the mentioned study. However, as for other crops, N is one of the most required nutrients for coffee, and leaf N levels and crop yield are positively related [40,41].

Although N is extremely mobile in the plant, it is one of the elements with the highest return rate to the soil (Figure 3). This may coincide with the complex dynamics of this element in the soil–plant system, related to fertilization applied in the area, to coffee flowering, or even due to the SOM content. Nitrogen is one of the elements with the highest content in cells and, even when recycled, a major portion still remains in senescent plant tissues [42]. 

The P 4.62–11 g kg^−1^ and S 1.33–2.7 g kg^−1^ contents of the coffee leaves were highest at the beginning and end of the evaluation periods. Between 90 and 180 days, a decline in P 2–4 g kg^−1^ and S 1.5 to 2 g kg^−1^ was observed and after 330 days, these contents increased again (Figure 3). At 180 days, the P content was highest for ‘IPR-103’ NB and ‘IPR-99’ NB, which coincides with the split fertilization of coffee, as similarly occurred in WB at 240 days. The S content was highest at 30 days in the ‘IPR-103’ NB treatment. Due to P fertilization in this study, the minimum and maximum levels were much higher than those reported elsewhere, which ranged from 0.39 to 2.2 g kg^−1^ [12,39].

The mean Ca content at the beginning of the evaluation was 11.9 to 13.5 g kg^−1^ and decreased throughout the evaluation period to 7 g kg^−1^ at 180 days. Thereafter, Ca increased from 240 days until the end of the evaluation period, at 360 days to 10 g kg^−1^ (Figure 3). At 60 days, the Ca content was highest in the ‘IPR-99’ NB system, at 150 and 210 days in the ‘IPR-103’ WB system, and at 240 days in the ‘IPR-103’ NB system. Calcium is the nutrient for which coffee beans have the third highest demand, after N and P, and it increases heat and water stress tolerance in plants [43]. The presence of *Urochloa decumbens* increased leaf nutrient contents of the macronutrient Ca at the end of the study period.

The potassium content remained constant in the ‘IPR-99’ NB system, while in the other production systems, it was highest at 30, 210, and 240 days (*p* < 0.05). This coincided with the resumption of irrigation, coffee flowering, and limestone application at 30 days; fruit maturation at 210 days; and the fruit harvest period at 240 days. Potassium deficiency causes a decrease in coffee yields, whereas an excess causes nutritional imbalance since it influences Ca and Mg uptake [44].

At the beginning of the evaluation period, the magnesium content was between 3.8 and 5.9 g kg^−1^, which declined after 120 days, at the moment of grain fruit formation, and increased at 240 days, coincidently with the coffee fruit harvest. At 30 days, the Mg content in the ‘IPR-99’ WB system was higher than in ‘IPR-103’ NB; at 150 days, the ‘IPR-103’ WB system had higher contents than the two cultivars without *Urochloa decumbens* intercropping; at 180 days, the Mg content was lower in the ‘IPR-99’ WB system compared to the others (*p* < 0.05). The higher concentration of Mg in coffee leaves in the first days of development during the first 150 days in the systems with *Urochloa decumbens* intercropped can be explained by the high effectiveness of *Urochloa* species in nutrient cycling. The species can accumulate 25 kg ha^−1^ of Mg in tissues [27]. The Mg contents observed in the different production systems were close to those reported by Salamanca-Jimenez et al. [40], who found levels between 2.46 and 8.30 mg kg^−1^. Magnesium contents in coffee are related to carbohydrate accumulation in leaves and roots [45].

In general, for macronutrients, P concentration was high, Mg, S, N, and Ca were in the ideal range of coffee tree demand, and K was low after 60 days, i.e., the period after water stress, when irrigation and/or rain returns and the crop is fertilized. These results are similar to those published by Maia and Conte [46], for the nutrients Mg 6.5 g kg^−1^, S 1.61 g kg^−1^, and Ca 14 g kg^−1^, but different for P 1.07 g kg^−1^, which decreased gradually in the treatments with *Urochloa decumbens*. The presence of *Urochloa decumbens* in intercrops must also have contributed to reducing K concentrations in the Arabica coffee cultivars (cvs ‘IPR-103’ and ‘IPR-99’). The results show that cultivars ‘IPR-99’ and ‘IPR-103’ are efficient in N, S, Mg, and Ca uptake, regardless of the intercropping with *Urochloa decumbens*.

During the fruit growth period, there is a decrease in macronutrients in the leaves, especially of K [12], and part of the nutrients are drained from the leaves to the grains, to provide grain filling, while Ca had an increased concentration, attributed in part to remobilization for the fruits.

The low K levels in the coffee plant (Figure 3) can be partly explained by the loss of free K, since the rapid cycling of K is related to the fact that this element is not part of any plant structure and appears in the ionic form K^+^ [47], which facilitates its release into the soil after the rupture of the plant tissues. The higher concentration of Ca was due to the participation of this element in the structural parts of the leaves and may be related to its low mobility and, therefore, slow decomposition [43].

In relation to the micronutrients, the model coefficients were significant for Cu in the intercropping system with *Urochloa decumbens* (WB) for both cultivars, in all systems for B (*p* < 0.05), and not significant for Mn and Fe (Figure 4). For Cu, the levels ranged from 2.24 to 7.68 mg kg^−1^ for NB and WB for both coffee cultivars. From 60 days onwards, with a mean of 7.21 mg kg^−1^ in all treatments (Figure 4), which is below the adequate content considered ideal for coffee plants, Cu levels increased and after 150 days, they decreased again. During the evaluation period, the Cu content did not differ between the production systems. After 120 days, the Cu concentration in leaves decreased in all treatments. This behavior can be attributed to the translocation of Cu from leaves to developing organs and fructification [48].

**Figure 3 plants-14-00496-f003:**
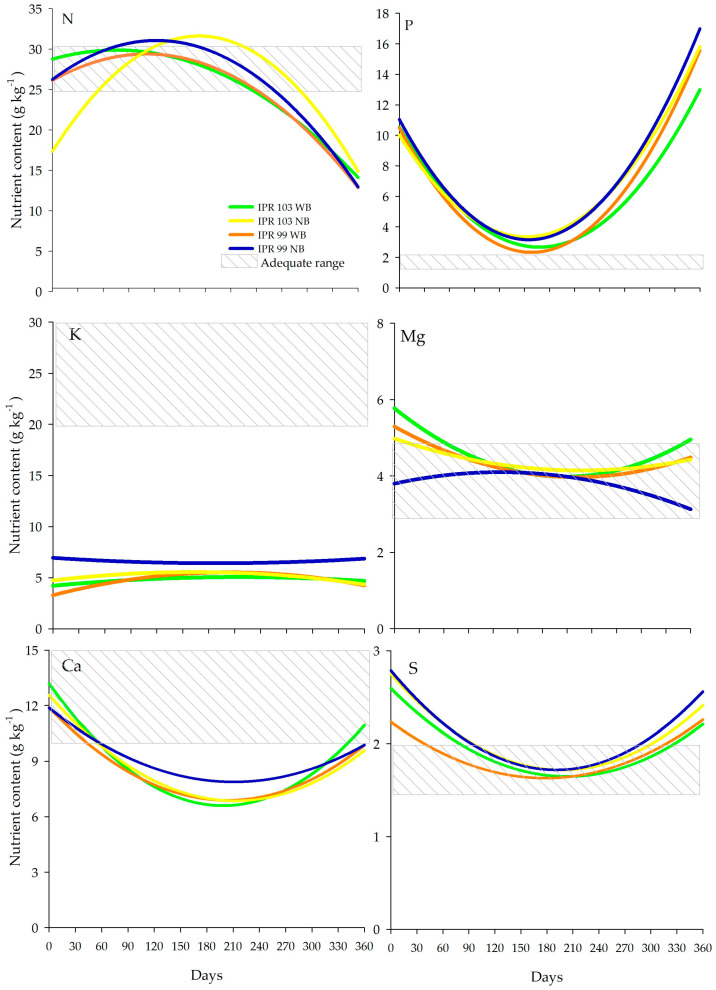
Macronutrient content in leaves and adequate range of two Arabica coffee cultivars, with and without *Urochloa decumbens* intercropping between coffee rows. ‘IPR-99’ WB, ‘IPR-103’ WB, ‘IPR-99’ NB, and ‘IPR-103’ NB as function of time. WB: with *Urochloa decumbens* intercropping and NB: without *Urochloa decumbens* between rows. Lines in each graph indicate adequate levels of nutrients; see Cantarella et al. and Malavolta [38,48].

The Mn content started in the range of 60–85 mg kg^−1^ and after 270 days of evaluation, it reached 28–50 mg kg^−1^. At 60 days, the Mn content of the ‘IPR-99’ WB system was higher than that of ‘IPR-103’ NB; at 180 days, in both cultivars without *Urochloa decumbens*, contents were higher than in the ‘IPR-99’ WB system (*p* < 0.05). At 270 days, the Mn content was higher in the ‘IPR-103’ WB than the ‘IPR-103’ NB system; and at 300 days, ‘IPR-99’ WB had a higher Mn content than ‘IPR-99’ NB (*p* < 0.05).

The B content showed no differences between the systems for any of the periods evaluated. For all the treatments, the highest B concentration in leaves occurred in the beginning of the development until the beginning of the fruit grain formation. The reduction in the concentration is due to the high demand in the reproductive phase (flowering and fructification) [12]. The Fe content at 60 days was higher in the ‘IPR-99’ NB system than ‘IPR-99’ WB and ‘IPR-103’ WB; after 150 days, it was higher in the ‘IPR-103’ WB system than the two cultivars without *Urochloa decumbens* (*p* < 0.05). The leaf nutrient concentrations peaked after 90 days, where B was 144.23; 131.67; 151.23; and 145.57 mg kg^−1^ and Fe was 563; 402.3; 401.1; and 410.1 mg kg^−1^, in the treatments ‘IPR-103’ WB; ‘IPR-99’ WB; ‘IPR-103’ NB; and ‘IPR-99’ NB, respectively.

Overall, the nutrients within the adequate range for coffee plants, as proposed by Cantarella [49], were N, Mg, Ca, S, B, Mn, and Fe, above the adequate range level P, and below this adequate range K and Cu. Knowledge of the nutritional status of the plant is fundamental, mainly in intercrops, as described in this study, for plant development, yields and the sustainability of the production systems.

Regarding the relationship between macronutrients in Arabica coffee leaves, the results for the N/S and K/Mg ratios were close to the levels considered ideal, according to Malavolta [48]. These ratios are important for productive growth and can be an information guide to ensure that nutrients are in balance.

Our results show that the introduction of *Urochloa decumbens* in the interrow spacing of the coffee plants were positive to nutrient cycling in the system, leading to better use efficiency of nutrients, especially Ca, Mg, Mn, and Fe in our experiment conditions. The Urochloa species have a vigorous and deep root that helps for recovery of nutrients which, probably, are out of the root zone of coffee plants. Then, the residue decomposition of the grass species provides nutrients to the main crop.

## 3. Materials and Methods

### 3.1. Experimental Location and Description

The study was carried out in the experimental area of Embrapa Cerrados (CPAC), (15°35′30″ S; 47°42′30″ W; 1200 m asl) in Planaltina-Distrito Federal, Brazil (Figure 5).

The soil in the area was classified as Typic Haplustox [50], and the soil chemical properties of the 0–10 and 10–20 cm layers are shown in Table 2. Particle analysis defined the mean levels of clay, silt, fine sand, and coarse sand as 601, 116, 47, and 236 g kg^−1^, respectively.

The climate was classified as tropical rainy (Aw), according to Köppen’s classification [51], with two well-defined seasons: rainy summers, from October to March, i.e., a rainy season, and dry winters, from April to September, corresponding to a dry period. Eighty percent of the rain falls in the rainy season between October and February. The mean annual air temperature is between 22 and 25 °C, and the mean annual rainfall is 1345.8 mm [45] (Figure 6).

Prior to the experiment, in the period from January 2000 to December 2007, Urochloa (synonym *Urochloa decumbens*) had been planted in the experimental area as a cover crop, without grazing. Thereafter, Arabica coffee cultivar Catuaí Vermelho 144 was planted and cultivated until 2016. After harvest, the coffee trees were severely pruned for renewal. 

In February 2019, two Arabica coffee cultivars (cvs. ‘I.P.R.103’ and ‘I.P.R.99’) were planted at a spacing of 3.5 m between rows and 0.5 m between plants. In the interrows without *Urochloa decumbens*, the herbicide Indaziflam was applied to control weeds under the coffee canopy at 180 mL/ha^−1^ with a backpack sprayer, and weeds were removed by hand. *Urochloa decumbens* was planted with a row spacing of 18 cm, at a seeding rate of 8 kg ha^−1^. In the plots with *Urochloa decumbens*, the forage was cut when it reached a height of 0.60 m and the plant residues were left on the soil surface. Prior to this experiment, a previous coffee plantation had been intercropped with *Urochloa decumbens*. Only at the beginning of this study, in February 2019, were the treatments separated in coffee monoculture and a coffee *Urochloa decumbens* intercrop. 

The experiment was arranged in randomized blocks in a factorial design with three replications. The first factor consisted of Arabica coffee intercropped with *Urochloa decumbens* (WB) and without intercropping (NB), and the second factor of two Arabica coffee cultivars (‘IPR103’ and ‘IPR99’).

Each plot comprised eight coffee trees. The plots were irrigated with a central pivot irrigation system and the irrigation management was based on the soil water content. When the soil moisture corresponded to a consumption of 50% of the available water in the 0–10 cm layer, the crop was irrigated. Water stress was applied for around 60 days between May and early September (dry season) to induce uniform flowering to increase productivity after resuming irrigation. The water content was monitored with ML1 (Delta-T Devices) moisture probes.

Coffee fertilization consisted of 300 kg ha^−1^ of triple superphosphate (41% P_2_O_5_). In the first year, side dressing fertilization consisted of 200 kg N ha^−1^ as urea and 200 kg K_2_O ha^−1^ as potassium chloride, applied in September, November, January, and March. In the second year, 200 kg N ha^−1^ as urea and 200 kg K_2_O ha^−1^ as potassium chloride were side-dressed four times (January, March, September, and November) with an added 50 kg ha^−1^ of fritted trace elements (FTE-BR 12). As of the third year, double the rates of the initial side dressing were applied four times (in September, November, January, and March) for N and K. Phosphorus was side dressed in the coffee rows, i.e., 2/3 of the total rate in September when irrigation was resumed and 1/3 between November and December. Liming consisted of 2 t ha^−1^ of dolomitic limestone (25% CaO + 25% MgO) and 2 t ha^−1^ agricultural gypsum (15% S + 16% Ca) after harvest in 2021.

### 3.2. Evaluation of Urochloa Plant Residues and Arabica Coffee Yield

The shoots of *Urochloa decumbens* were cut 10 times, from October 2021 to February 2023 (Table 3). The chemical composition of the plants at the 3rd cut was analyzed. 

For *Urochloa decumbens* shoot sampling, 0.50 × 0.50 m square frames were randomly laid in the center of each plot, with two repetitions per plot. The plant material in the frames was cut, weighed, and dried to a constant weight in a forced ventilation oven at 65 °C. Thereafter, the material was re-weighed to determine dry matter. For the analysis of *Urochloa decumbens* straw (3rd cut), the different samples were grounded and the cellulose, hemicellulose, and lignin content of the cover species was determined using NIRS FOSS 5000, System II type 461,006 (FOSS Analytical SA, DK 3400 Hilleroed, Denmark) with ISIScan v.2.85.3 software (ISI Software, FOSS Analytical AB, Höganãs, Sweden) and Unscrambler X 10.5.1 software (CAMO Software AS, Oslo, Norway).

The nutrients were extracted by digestion with perchloric acid and hydrogenperoxide, heated to 350 °C in a digester block for one hour, and analyzed by plasma emission spectrophotometry. For nutrient analysis, 2 g of the ground samples were filled in digestion tubes with extractor solution composed of perchloric acid and hydrogenperoxide, for around 1 h in a digestion tube. After the material was digested, Milli-Q water was added to complete a volume of 50 mL. The elements Mg, K, Ca, P, S, Fe, Mn, Cu, and B were analyzed in optical plasma (ICP OES), using the following lines (nm): Ca 422.6, K 769.8, Mg 285.2, P 178.2, S 180.7, Fe 238.2, Mn 257.6, Cu 324.7, and B 208.8.

The digestion for total nitrogen was performed with perchloric acid and hydrogen peroxide, heated to 350 °C in a digester block for one hour, and analyzed colorimetrically on a Lachat QuikChem 8500 autoanalyzer, using a series 2 flow-injection analyzer. The content of the digestion vessels was transferred to reading tubes, diluted at 1:6 with Milli-Q water.

The Arabica coffee bean yield was determined by harvesting three plants in the center of each plot, and after drying and weighing, the data were converted into bags ha^−1^ (one bag represents 60 kg).

### 3.3. Determination of Macro- and Micronutrient Concentrations in the Shoots of Two Arabica Coffee cvs. ‘IPR-103’ and ‘IPR-99’, in Different Crop Stages

Arabica coffee leaves were collected from two cultivars (‘IPR-103’ and ‘IPR-99’) with (WB) and without (NB) *Urochloa decumbens* intercropping. Leaf samples were collected from the middle third of the plants at the end of the dry period, from September 2021 to September 2022, in 30 day intervals, over a 12-month period; that is, a total of 13 samplings (after 0, 30, 60, 90, 120, 150, 180, 210, 240, 270, 300, 330, 360 days).

The coffee stages in which leaf nutrient analyses were performed are described below: Phases of the phenological cycle of coffee: first collection of coffee leaves in September 2021, at the end of the period of water stress (time 0). In October (30 days), irrigation was resumed, with active leaf development and leaf bud formation. In November (60 days), the end of coffee flowering is observed, which occurs between September and November. Flowering is generally concentrated, resulting in uniform fruit ripening. In December (90 days), fertilization occurs in split applications, at the beginning of fruit formation and grain filling (Figure 7).

In January (120 days), fruit grain formation takes place from January to March, involving continuous fruit development and the beginning of fruit filling. February (150 days): fruit expansion and nutrient accumulation. March (180 days): the continuation of fruit filling for maturation. April (210 days): fruit color change, indicating physiological maturity. May (240 days): coffee harvest at ideal fruit maturity. The fruits change from green to red or yellow, and harvest can last for weeks. June (270 days): the continuation of the harvest of ripe fruits. July (300 days): trees enter dormancy after harvest, with the senescence of non-productive branches. August (330 days): the preparation for the next cycle with the growth of new leaves; the last coffee harvest occurred in September 2022 (360 days) (Figure 7). 

Leaves were taken from the middle third of coffee trees, randomly selected in the plot, from the third to the fourth pair of leaves on the branches. The samples were cleaned and stored in paper bags and oven-dried at 65 °C for 72 h. The samples were ground and the material was digested. For macronutrient analysis, 2 mg of the ground samples were filled in digestion tubes, and 10 mL of an extraction solution (5 mL perchloric acid, 4 mL Milli-Q water, and 1 mL hydrogen peroxide) were used per 100 mg of sample for 1 h in a digestion block at a digestion temperature of 350 °C. After the material had been digested, Milli-Q water was added to complete the volume to 50 mL. Macro- and micronutrient analysis was performed by inductively coupled plasma emission spectroscopy (ICP-OES, using the following lines (nm): Ca 422.6, K 769.8, Mg 285.2, P 178.2, S 180.7, Fe 238.2, Mn 257.6, Cu 324.7, and B 208.8). In addition, total nitrogen was determined and digestion analyzed with perchloric acid and hydrogen peroxide, then heated to 350 °C in a digester block for one hour and analyzed colorimetrically on a Lachat QuikChem 8500 autoanalyzer, using a series 2 flow-injection analyzer. The content of the digestion vessels was transferred to reading tubes, diluted at 1:6 with Milli-Q water.

### 3.4. Data Processing and Analysis

The function used for the nutrient contents (N, Ca, K, P, Mg, S, Cu, Mn, Fe, and B) for both treatments studied (with and without *Urochloa decumbens* intercropping between coffee rows) was the polynomial fitting model.

The data of nutrient levels over the course of time, for the total dry matter in cuts of *Urochloa decumbens* intercropped, and for the yield of two Arabica coffee cultivars with and without intercropping with *Urochloa decumbens* were tested for normality of distribution (Shapiro–Wilk) and then analysis of variance (ANOVA) was performed. The different sampling periods were considered categorical variables, and the means were compared by Tukey’s test (*p* < 0.05) using the software R (version 3.2.2).

## 4. Conclusions

The ‘IPR-99’ cultivar intercropped with *Urochloa decumbens* achieved 400 kg ha^−1^ (8 bags) more than the other treatments, with a consequent production cost reduction. The intercropping systems can lead to improvement in Arabica coffee yield.

The *Urochloa decumbens* leaf nutrients were on the adequate range or even above the ideal, showing efficiency in nutrient cycling. Increased levels of the macronutrients Ca and Mg and micronutrients Mn and Fe were confirmed in Arabica coffee leaves with *Urochloa decumbens* intercropped, highlighting the contribution of the grass species to these nutrients’ recovery. For all nutrients, the concentration in leaves varied conforming to Arabica coffee phenological phases and the demand of nutrients for flowering and fruit formation. In these phases, the overall leaf content decreased because of the nutrient remobilization, not depending on the management system.

The assessments of nutrient content in leaves show that *Urochloa decumbens* in the interrow spacing of the Arabica coffee plants functioned as an efficiency management system as regards nutrient cycling and the improvement of nutrient uptake for Ca, Mg, Mn, and Fe for the main crop in these experiment conditions. Direct benefits of the introduction of *Urochloa decumbens* also may have affected the Arabica coffee plants, such as in soil protection and the better use of environmental resources, such as water.

We suggest the evaluation of long-term experiments with Arabica coffee intercropped with *Urochloa decumbens* in more than one season to improve the understanding of the nutrient uptake by the main crop.

## Figures and Tables

**Figure 1 plants-14-00496-f001:**
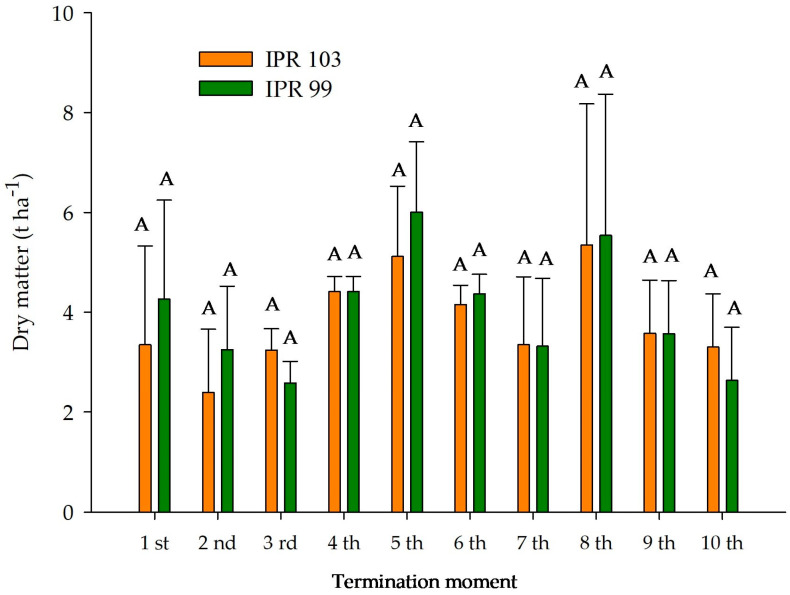
Total dry matter in cuts of *Urochloa decumbens* intercropped with two Arabica coffee cultivars (cvs ‘IPR-103’ and ‘IPR-99’). Same letters do not differ from each other by Tukey test at 5% probability.

**Figure 2 plants-14-00496-f002:**
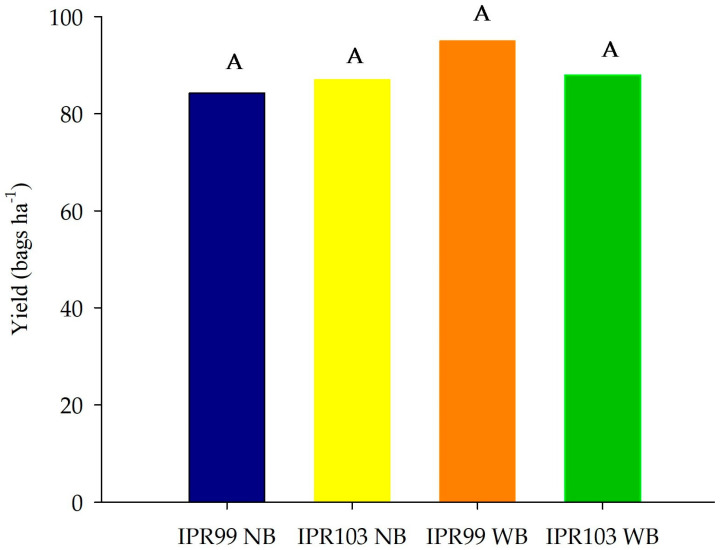
Yield of two Arabica coffee cultivars with and without intercropping with *Urochloa decumbens* in 2021 in experimental area of Embrapa Cerrados, Planaltina-DF. WB: with Brachiaria (*Urochloa decumbens*) intercropping between coffee rows and NB: without Brachiaria (*Urochloa decumbens*) between coffee rows. Same letters do not differ from each other by Tukey test at 5% probability for each treatment.

**Figure 4 plants-14-00496-f004:**
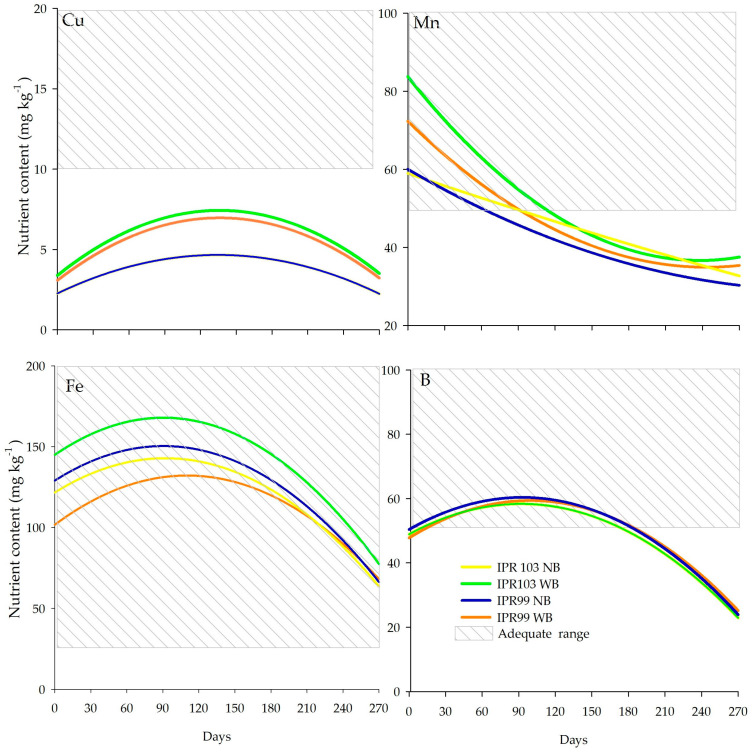
Micronutrient content in the leaves and the adequate range of two Arabica coffee cultivars, with and without intercropping with *Urochloa decumbens* between the rows. ‘IPR-99’ WB, ‘IPR-103’ WB, ‘IPR-99’ NB, and ‘IPR-103’ NB as a function of time. WB: with *Urochloa decumbens* intercropping and NB: without *Urochloa decumbens* between the rows. Lines in each graph indicate adequate levels of nutrients; see Cantarella et al. and Malavolta [38,48].

**Figure 5 plants-14-00496-f005:**
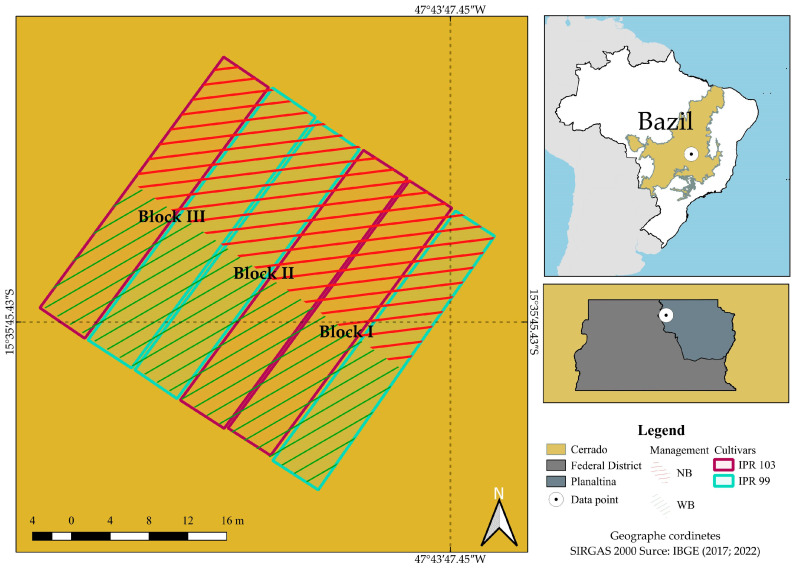
The experimental area at Embrapa Cerrados in the Cerrado biome Planaltina, DF. Arabica coffee with and without *Urochloa decumbens* intercropped in the coffee interrows.

**Figure 6 plants-14-00496-f006:**
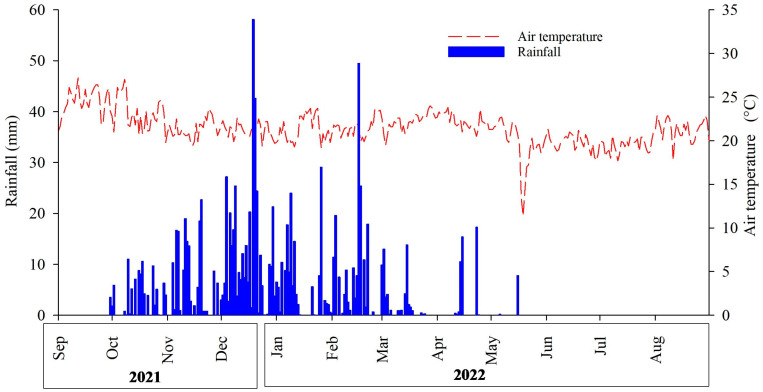
Mean air temperature (°C) and mean rainfall (mm) in the experimental area of Embrapa Cerrados from September 2021 to August 2022 in Planaltina, DF.

**Figure 7 plants-14-00496-f007:**
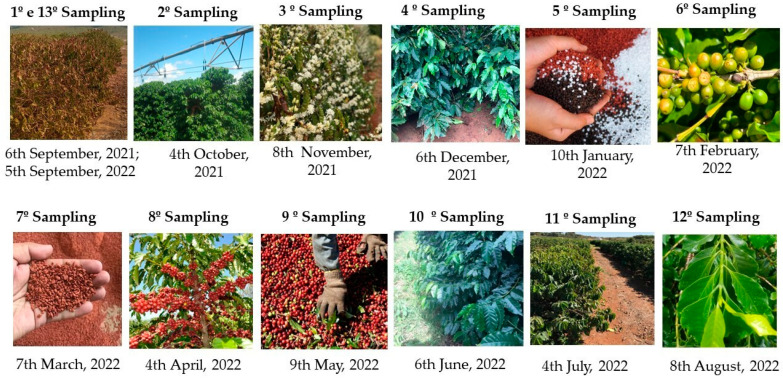
Monthly samplings that represent all Arabic coffee phases, from September 2021 to August 2022, in the experimental area of Embrapa Cerrados in Planaltina, DF.

**Table 1 plants-14-00496-t001:** Chemical composition and concentrations of macro- and micronutrients of *Urochloa decumbens* residues after intercropping with Arabica coffee.

Hemicellulose	g kg^−1^	263.9 (±10.9)
Cellulose		314.8 (±10.1)
Lignin		30.6 (±22)
Lignin/N		1.6 (±0.02)
N	g kg^−1^	18.96 (±2.92)
P		6.5 (±0.42)
K		27.9 (±2.73)
Ca		4.9 (±0.76)
Mg		4.4 (±0.66)
S		1.8 (±0.12)
B		78.5 (±6.85)
Cu	mg kg^−1^	5.0 (±0.44)
Fe		1602.3 (±1103)
Mn		30.5 (±7.66)

**Table 2 plants-14-00496-t002:** The soil chemical and physical properties of the experimental area under management systems. Coffee cultivars ‘IPR-103’ and ‘IPR-99’. WB (Arabica coffee intercropped with *Urochloa decumbens*) and NB (Arabica coffee without *Urochloa decumbens*).

		IPR103WB	IPR99WB	IPR103 NB	IPR99NB	IPR103 WB	IPR99 WB	IPR103 NB	IPR99NB
Property	Measurement Unit	Soil Depth 0–10 cm				Soil Depth 10–20 cm			
Organic matter	mg kg^–1^	34	35	32	33	26	27	28	26
pH	(H_2_O)	5.3	5.3	5.2	5.3	5.6	5.6	5.5	5.6
P	mg dm^–3^	13.9	14.2	25.5	23.3	11.7	13.2	18.3	16.0
H + Al	mg dm^–3^	7.4	7.8	7.7	7.2	6.5	6.4	5.7	5.5
SB	mg dm^–3^	7.1	7.4	7.5	8.1	6.5	6,7	7.0	6.8
CEC	mg dm^–3^	7.3	7.6	7.6	8.2	6.6	6.8	7.1	6.8
V	%	49	49	49	52	50	51	54	55
Soil density	g cm^–3^	1.39	1.39	1.4	1.4	1.36	1.36	1.36	1.36
Al	mg dm^–3^	0.14	0.19	0.11	0.06	0.04	0.07	0.03	0.03
Ca	mg dm^–3^	4.38	4.54	4.70	5.18	4.56	4.66	4.85	4.61
Cu	mg dm^–3^	4.66	4.78	4.93	5.54	2.15	2.37	2.58	2.47
K	mg dm^–3^	447	504	437	489	194	208	240	272
Mg	mg dm^–3^	1.63	1.65	1.68	1.70	1.53	1.57	1.58	1.52
Mn	mg dm^–3^	14.0	13.2	11.9	11.3	4.22	4.09	4.13	3.97

P = phosphor; H + Al = potential acidity; SB = sum of bases; CEC = cationic exchange capacity; V = base saturation; Al = aluminum; Ca = calcium; Cu = copper; K = potassium; Mg = magnesium; Mn = manganese. pH in water at a soil:solution ratio of 1:1; Al, Ca, and Mg extracted by KCl 1 mol L^−1^; P and K extracted by Mehlich^−1^ method; cation exchange capacity determined using pH 7.0 buffered ammonium acetate solution.

**Table 3 plants-14-00496-t003:** Cutting times of *Urochloa decumbens* intercropped with Arabica coffee at Embrapa Cerrados Planaltina, DF.

*Urochloa decumbens* Cuttings	
1st Cutting	Accessed on 19 October 2021
2nd Cutting	Accessed on 16 November 2021
3rd Cutting	Accessed on 13 January 2022
4th Cutting	Accessed on 22 February 2022
5th Cutting	Accessed on 30 March 2022
6th Cutting	Accessed on 18 July 2022
7th Cutting	Accessed on 18 November 2022
8th Cutting	Accessed on 14 December 2022
9th Cutting	Accessed on 12 January 2023
10th Cutting	Accessed on 14 February 2023

## Data Availability

The data presented in this study are available on request from the corresponding authors.

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
