# Peer review of "Arabica coffee Intercropped with Urochloa decumbens Improved Nutrient Uptake and Yield in the Brazilian Cerrado"

_plants, 2025, doi:10.3390/plants14040496_

Round 1
Reviewer 1 Report
Comments and Suggestions for Authors
Arabica coffee is one of the most important cash crop in Brazil, the study on yield and quality of Arabica coffee by intercropping with Urochloa decumbens has a great importance. This manuscript reported that the nutrient uptake and yield of Arabica coffee by intercropping with Urochloa decumbens in the Brazilian Cerrado. It is important to explore the effective approach to improve yield and quality of Arabica coffee. However, I think the following comments should be considered.
(1) TITLE: The title should be converted to an informative title from an indicative title, such as "Arabica coffee intercropping with Urochloa decumbens improved nutrient uptake and yield in the Brazilian Cerrado".
(2) ABSTRACT: The research GAP of this research should be clearly stated in the ABSTRACT.
(3) INTRODUCTION: The research GAP of this research should be clearly stated in the INTRODUCTION.
(4) RESULTS AND DISCUSSION: Figures 1 and 2 have a similar color, which should be improved using a showy color. Also, what meaning of the letter "A" should be explained in figure legend or METHODS. Figure 2 is the same as figure 1.
(5) METHODS: ok.
(6) CONCLUSION: OK.
(7) REFERENCES: OK.
Author Response
January 10, 2025
Embrapa Cerrados
BR-020, km 18, Planaltina
73310-970, DF, Brazil
Ms. Lea Tao
Assistant Editor
MDPI Plants Editorial Office
Grosspeteranlage 5, 4052 Basel, Switzerland
Dear Lea Tao,
We thank the reviewers and editor for their relevant and complete contributions and their thoughtful and valuable comments to improve our submitted manuscript.
We made an effort to respond to all requests from the three reviewers and the editor.
We reformulated the research manuscript sections as requested by the reviewers.
We corrected every point of the manuscript in Office/Word modification mode. We have included here the reviewers’ comments and how we have addressed each point, in italics the reviewers’comments and our response in red.
- – Reviewer 1 –
Title:
The title should be converted to an informative title from an indicative title, such as "Arabica coffee intercropping with Urochloa decumbens improved nutrient uptake and yield in the Brazilian Cerrado".
The title was converted to Arabica coffee intercropped with Urochloa decumbens improved nutrient uptake and yield in the Brazilian Cerrado.
Abstract:
The research GAP of this research should be clearly stated in the ABSTRACT.
The abstract was rewritted to attend the suggestion.
Introduction:
The research GAP of this research should be clearly stated in the INTRODUCTION.
The introduction was revised and reformulated and some paragraphs were improved to better understanding the importance of integrated systems in nutrient cycling.
Results and discussion:
Figures 1 and 2 have a similar color, which should be improved using a showy color. Also, what meaning of the letter "A" should be explained in figure legend or METHODS. Figure 2 is the same as figure 1.
The figures were edited as requested. The explanation about the letters was insert in the legend.
We appreciate it and would be glad to respond to any further questions and comments you may have.
On behalf of all authors. Sincerely yours,
Ms. Thais Rodrigues de Sousa
Reviewer 2 Report
Comments and Suggestions for Authors
General comments: the manuscript investigated the biomass production, nutrient contents and yield of a cover grass species with two coffee varieties (cvs. IPR-103 and IPR-99) in Brazil. The aims were to evaluate the leaf nutrient content and yield of two cultivars intercropped or not with Urochloa decumbens under irrigation conditions. The writing needs improvement, and some descriptions was too tedious and can be more concise, especailly in the materials and methods. The results description order should be modified, and the order should be cover plant biomass production, and then nutrient back to field. The data analysis was not enough, and how about coffee yield and quality? The simulated equations can be collected using a supplementary table, which was easier to compare and read. The illustration in 3.3 can be combined to more clearly know when and where to sample the leaves. The experiment was only carried on for growth season, and suggest add a "Limitation" section to show the shortcoming and some needed further investigation about this study. Some specific comments were as following:
Title: title was not accurate.
Abstract: it did not fully cover the main contents of the manuscript, such as the detailed treatments and the key results, and a summarized conclusion was also lacking.
Key words: which should be scientific terminologies.
Introduction: the illustration logicality needs improvement, and there exist statement jump and too many short paragraphs.
Ø Line 44: the citation mark was not correct.
Ø Line 52: not clear.
Ø Lin 58-59: more detailed information needed.
Ø Line 74: why choose this two varieties?
Results and Discussion: should present the results directly and then explain the reasons or mechanisms behind.
Ø Line 102-103: not clear.
Ø Line 119: figure 1, the legend did not show the content of the column figure, and statistical analysis mark above the columns were absent.
Ø Line 136: the non-significant results may be due to one year experiment.
Ø Line 141-146: no data results supporting.
Ø Line 148: the section title was too long.
Ø Line 258: the figures can be modified to put the index as Y-axis.
Ø Line 289: not clear and should be marked in the left photo.
Ø Line 297: tables and figures should be self-explanatory.
Ø Line 297: table 2 the unit should be SI, and the experiment was not clear.
Ø Line 318: what was Indaziflan? More information needed.
Ø Line 329: the abbreviations should be unified across the manuscript.
Ø Line 332-336: more description needed for the irrigation and water stress, when and why?
Ø Line 351-353: how much biomass was cut and back to the field for 10 times? Why only the 3rd time was analyzed?
Ø Line 360 Table 360: the table can be combined into line 351 as described directly.
Ø Line 363-365: what was this measurement for?
Ø Line 378: here the plant was taken cover crop, why measure their forage quality?
Ø Line 394: SI needed, how much g or kg for one bag?
Ø Line 399: in which direction?
Ø Line 422: the year time should be pointed out after the month.
Ø Line 439: how about differences between treatments, such as varieties, intercropping or not?
Conclusion: the results were not fully and comprehensively summarized according to the resutls and discussion, and lacks of a specific conclusion.
Ø Line 449-450: can be deleted.
Ø Line 458-460: meaningless and can be deleted.
Comments on the Quality of English LanguageThe concise and logicality need improvement.
Author Response
January 10, 2025
Embrapa Cerrados
BR-020, km 18, Planaltina
73310-970, DF, Brazil
Ms. Lea Tao
Assistant Editor
MDPI Plants Editorial Office
Grosspeteranlage 5, 4052 Basel, Switzerland
Dear Lea Tao,
We thank the reviewers and editor for their relevant and complete contributions and their thoughtful and valuable comments to improve our submitted manuscript.
We made an effort to respond to all requests from the three reviewers and the editor.
We reformulated the research manuscript sections as requested by the reviewers.
We corrected every point of the manuscript in Office/Word modification mode. We have included here the reviewers’ comments and how we have addressed each point, in italics the reviewers’comments and our response in red.
- – Reviewer 2 –
General comments:
The manuscript investigated the biomass production, nutrient contents and yield of a cover grass species with two coffee varieties (cvs. IPR-103 and IPR-99) in Brazil. The aims were to evaluate the leaf nutrient content and yield of two cultivars intercropped or not with Urochloa decumbens under irrigation conditions. The writing needs improvement, and some descriptions were too tedious and can be more concise, especially in the material and methods.
Some parts of the manuscript were edited, especially in the Material and Methods to attend this suggestion.
The results description order should be modified, and the order should be cover plant biomass production, and then nutrient back to field.
The order was modified.
The data analysis was not enough, and how about coffee yield and quality? The simulated equations can be collected using a supplementary table, which was easier to compare and read. The illustration in 3.3 can be combined to more clearly know when and where to sample the leaves. The experiment was only carried on for growth season, and suggest add a "Limitation" section to show the shortcoming and some needed further investigation about this study.
Title:
Title was not accurate.
The title was converted to Arabica coffee intercropped with Urochloa decumbens improved nutrient uptake and yield in the Brazilian Cerrado.
Abstract:
It did not fully cover the main contents of the manuscript, such as the detailed treatments and the key results, and a summarized conclusion was also lacking.
The abstract was rewritted to attend the suggestion.
Key words:
Which should be scientific terminologies.
The key words were edited for attend the suggestion.
Introduction:
The illustration logicality needs improvement, and there exist statement jump and too many short paragraphs.
The introduction was revised and reformulated and some paragraphs were improved to better understanding the importance of integrated systems in nutrient cycling.
Ø Line 52: not clear.
The part of the text was edited to better comprehension.
Ø Lin 58-59: more detailed information needed.
The paragraph was improved.
Ø Line 74: why choose this two varieties?
The varieties studied are the most used in Brazilian Cerrado by producers and have similar plant height avoiding competition when planted beside.
Results an discussion:
Should present the results directly and then explain the reasons or mechanisms behind.
The text was edited for better comprehension.
Ø Line 102-103: not clear.
This part was deleted.
Ø Line 119: figure 1, the legend did not show the content of the column figure, and statistical analysis mark above the columns were absent.
The figure has been replaced and the legend was improved.
Ø Line 141-146: no data results supporting.
The references of the sentence supports the affirmation.
Ø Line 148: the section title was too long.
The section title was edited to a shorter one.
Ø Line 258: the figures can be modified to put the index as Y-axis.
The Y-axis was added to the figure.
Ø Line 297: tables and figures should be self-explanatory.
Tables and figures were edited for better understanding.
Ø Line 297: table 2 the unit should be SI, and the experiment was not clear.
The unit was corrected for the SI unit.
Ø Line 318: what was Indaziflan? More information needed.
A herbicide to control the incidence of weeds.
Ø Line 329: the abbreviations should be unified across the manuscript.
The hole text has the same abbreviations after revision.
Ø Line 332-336: more description needed for the irrigation and water stress, when and why?
More informations about irrigation was added.
Ø Line 351-353: how much biomass was cut and back to the field for 10 times? Why only the 3rd time was analyzed?
The Urochloa decumbens was mowed and the grass straw was left in the soil surface (~5t/ha). The objective of this analyze was only to characterize the nutrient content in grass leaves.
Ø Line 363-365: what was this measurement for?
The measurement was performed to characterize the nutrient content in grass leaves.
Ø Line 378: here the plant was taken cover crop, why measure their forage quality?
The hemicellulose, cellulose and lignin content does not express only the quality of forage for animal nutrition but it’s also related to plant residues decomposition and consequently the release of nutrients.
Ø Line 394: SI needed, how much g or kg for one bag?
The information was added in the manuscript.
Ø Line 399: in which direction?
The information was added in the manuscript.
Ø Line 422: the year time should be pointed out after the month.
The information was added in the manuscript.
Ø Line 439: how about differences between treatments, such as varieties, intercropping or not?
The statistical analysis performed was described in the manuscript.
Conclusions:
The results were not fully and comprehensively summarized according to the results and discussion, and lacks of a specific conclusion.
The conclusion was rewritted for better understanding.
Ø Line 449-450: can be deleted.
The conclusion was rewritted for better understanding.
Ø Line 458-460: meaningless and can be deleted.
The conclusion was rewritted for better understanding.
We appreciate it and would be glad to respond to any further questions and comments you may have.
On behalf of all authors. Sincerely yours,
Ms. Thais Rodrigues de Sousa
Reviewer 3 Report
Comments and Suggestions for Authors
Recommendations and questions
Abstract
First sentence -a little unclear, whether this means - Intercropping coffee lines with cover crops such as Urochloa decumbens can increase its productivity due to higher exudation and efficient nutrient cycling?
The purpose of the work is not clearly stated. Urochloa decumbens intercropped with coffee or vice versa? Or does it not matter? Crop productivity – coffee or Urochloa decumbens? It is necessary to formulate it clearly and precisely. This also applies to the concluding part of the Introduction.
Line 25-26. “Despite a positive difference of 8 bags ha-1 in cultivar 'IPR-99' with Urochloa decumbens between the rows, there was no significant difference in yield of the coffee cultivars in the treatments with and without Urochloa decumbens intercropped between the rows.” Figure 2 shows completely different results - the highest yield was obtained without intercropping. So where is the mistake? Is such a conclusion even correct?
Results and Discussion
Line 98-99. “The macronutrient levels were within the standard considered adequate for Urochloa decumbens”. It would be very valuable if you could specify this standard range, either in text or as a separate column in a table.
Line 108-110. “The concentrations of the micronutrients B, Fe and Mn were considered below the ideal critical range for Urochloa decumbens.” Same question as above, especially considering that the specified micronutrient concentrations are high enough for almost any plant.
Line 122-124. “Despite a positive difference of 8 bags ha-1 in cultivar 'IPR-99' with Urochloa decumbens between the rows, there was no significant difference in relation to the yield of the coffee cultivars in the treatments with and without Urochloa decumbens intercropped between the rows (Figure 2)”. Figure 2 shows completely different results - the highest yield was obtained without intercropping!!!
2.2 Leaf nutrient content of Arabica coffee cultivars with and without Urochloa decumbens intercropping
Macronutrients
Line 160-161. “The coffee N contents varied over time, from 8 to 26 g kg-1 at the beginning of the evaluation (water-stressed environment) to 32 to 38 g kg-1, after 60 days…” Really? The Figure 3 shows that the lowest value could be ~ 18 g/kg.
In general, it should be clearly stated whether intercropping improved the supply of the coffee trees with the studied nutrient or no.
Also, it should be clearly defined whether the content of the nutrient, for example S, was sufficient during the active vegetation period. This is not really clear from the text. Also, regarding nutrients, it would be very informative to provide certain limits of sufficiency. Either in the text or by marking them in the Figures.
In Abstract: The presence of Urochloa decumbens increased leaf nutrient contents: of the macronutrients Ca and Mg and micronutrients Mn and Fe. How can we understand from the text devoted to the Ca content in the leaves (Line 189-195) that one option was better than another?
Line 199-201. “Potassium deficiency causes a decrease in coffee yields whereas an excess causes nutritional imbalance, since it influences Ca and Mg uptake [43].” Why is there talk about K deficiency here? Was there a K deficiency in the coffee leaves? Why is nothing clearly described and compared to standard values? The reader should not have to guess what the author wanted to say. A scientific article should be based on facts, after all.
Although the Mg content in leaves has been more or less clearly described, no conclusion has been given about the impact of intercropping.
Micronutrients.
Line 233-234. “For Cu, the levels ranged from 0 to 7.68 mg kg-1 for NB and 0.7 to 18 mg kg-1 for WB for both coffee cultivars.” It is not possible for the Cu content to be 0 mg/kg. Since Cu is an essential nutrient, the plant must die. Rather, the detection method was not sensitive enough, or the sample weight was too small.
“which is below the critical content considered ideal for coffee plants…” And again, you don't specify this Cu value. This is annoying. Is there really some special secret - the optimal concentrations of nutrients in coffee tree leaves?
Line 245-246. “The B and Fe contents were evaluated over 12 months (Figure 4). There were no differences between the systems in any of the periods evaluated.” Didn't the other nutrients have a 12-month period?
Here again I don't understand - if there were no differences in Fe, how can you write in your conclusions that intercropping improved Fe supply?
Line 254-255. “The high mobility of B is due to its presence in the soil in the form of a nonionized molecule. As a result, Zn is transported in the soil by diffusion while B is made available to plants.” Now everything is in a mess! What does Zn have to do with boron untake?
Legends to Figure 3 and 4. “Macronutrient concentrations in two Arabica coffee cultivars…” Since coffee leaves have been analysed - it should be indicated - in leaves.
Although at the end of the chapter nutrients are named whose supply is sufficient, low, high; nothing is mentioned about the results of the experiment with Urochloa decumbens intercropping - is this technology effective or not.
Materials and Methods
This is one of the most important sections in any scientific article, so it requires special accuracy and clarity.
3.1 Experimental location and description
Table 2. Soil chemical and physical properties of the experimental area under management systems.
All Figures and Tables should be self-explanatory. The legends should indicate the meaning of all used symbols and abbreviations. What is IPR103WB, IPR99WB etc.? If P is determined in Mehlich extraction, in what other nutrients? Why did you explain that B = boron; Zn = zinc, if it is a well-known fact, but the most incomprehensible thing is that these nutrients have not been analysed at all!
This gives the impression of a certain carelessness of the authors.
The setup of the experiment in relation to previous coffee cultivation is described in a rather incomprehensible way.
Line 333-335. Why references? Clearly distinguish what you have done from other research results. Did the mentioned authors do watering or no watering in your experimental field?
Were the fertilizer doses divided into 4 parts, or was the specified dose applied each month?
3.2 Evaluation of Urochloa plant residues and Arabica coffee yield
Line 383-385. “For nutrient analysis, 2 mg of the ground samples were filled in digestion tubes with 10 ml of extractor solution composed of 5 mL nitric acid, 4 mL of Milli-Q water and 1 mL of hydrogen peroxide, for each 100 mg of sample, for around 2 h in a digestion tube.” Absolutely illogical and unbelievable! How could you weigh 2 mg? Why so little? It is no wonder then that the Cu content could not be determined, since the same methodology is also specified for coffee leaf analysis!
Line 390-392. “To determine total nitrogen, the content of the digestion vessels was transferred to reading tubes, diluted at 1:6 with Milli-Q water, analyzed colorimetrically on a Lachat QuikChem 8500 autoanalyzer, using a series 2 flow-injection analyzer.” Also illogical and unbelievable. How were you able to determine the N content if you previously wrote that the sample was mineralized using nitric acid? These same questions apply to coffee leaf analyses.
Conclusion
The conclusions should give answers to the purpose of the research and the questions rose in the tasks.
“The objective of this study was to evaluate the effect of Urochloa decumbens intercropped with two coffee cultivars (Coffea arabica L.) on the levels of macro and micronutrients and crop productivity.”
or
“The authors of this study hypothesized that in the Brazilian Cerrado, the inclusion of Urochloa decumbens intercropped with Arabica coffee would increase leaf nutrient contents and coffee yield. To this end, we evaluated the leaf nutrient content and yield of two Arabica coffee cultivars (cvs. IPR-103 and IPR-99) intercropped or not with Urochloa decumbens, under irrigation, in the Brazilian Cerrado.”
I rekomend to rewrite the conclusions, giving a clear and research-based answer to the raised questions.
Also, conclusions should provide information, a conclusion, hypothesis on how the knowledge gained in this study could be useful in the future, in practice, in creating growing technologies.
Concluding remarks.
Overall, the article leaves a dual impression. On the one hand, the topic is relevant today, the article contains new knowledge and quite extensive data material. On the other hand, many inaccuracies, incomplete information, inconclusive evaluation of results, errors in methodological descriptions. There is no real certainty about statistically significant differences between the treatments. At the moment there are more questions than answers.
Therefore, it is necessary to make significant corrections: clearly describe the methodological part, purposefully improve the Results and Discussion, critically analysing the obtained data.
I recommend accepting this article in Plants after major revision.
Relatively broad suggestions and recommendations for improving the quality of the article are provided in the comments to the authors.
Author Response
January 10, 2025
Embrapa Cerrados
BR-020, km 18, Planaltina
73310-970, DF, Brazil
Ms. Lea Tao
Assistant Editor
MDPI Plants Editorial Office
Grosspeteranlage 5, 4052 Basel, Switzerland
Dear Lea Tao,
We thank the reviewers and editor for their relevant and complete contributions and their thoughtful and valuable comments to improve our submitted manuscript.
We made an effort to respond to all requests from the three reviewers and the editor.
We reformulated the research manuscript sections as requested by the reviewers.
We corrected every point of the manuscript in Office/Word modification mode. We have included here the reviewers’ comments and how we have addressed each point, in italics the reviewers’comments and our response in red.
- – Reviewer 3 –
Abstract:
First sentence - a little unclear, whether this means - Intercropping coffee lines with cover crops such as Urochloa decumbens can increase its productivity due to higher exudation and efficient nutrient cycling?
The abstract was rewritted to attend the suggestion.
The purpose of the work is not clearly stated. Urochloa decumbens intercropped with coffee or vice versa? Or does it not matter? Crop productivity – coffee or Urochloa decumbens? It is necessary to formulate it clearly and precisely. This also applies to the concluding part of the Introduction.
The abstract was rewritted to attend the suggestion.
Line 25-26. “Despite a positive difference of 8 bags ha-1 in cultivar 'IPR-99' with Urochloa decumbens between the rows, there was no significant difference in yield of the coffee cultivars in the treatments with and without Urochloa decumbens intercropped between the rows.” Figure 2 shows completely different results - the highest yield was obtained without intercropping. So where is the mistake? Is such a conclusion even correct?
The abstract was rewritted to attend the suggestion.
In Abstract: The presence of Urochloa decumbens increased leaf nutrient contents: of the macronutrients Ca and Mg and micronutrients Mn and Fe. How can we understand from the text devoted to the Ca content in the leaves (Line 189-195) that one option was better than another?
The abstract was rewritted to attend the suggestion.
Results and discussion:
Line 98-99. “The macronutrient levels were within the standard considered adequate for Urochloa decumbens”. It would be very valuable if you could specify this standard range, either in text or as a separate column in a table.
The standard range was included in the paragraph.
Line 108-110. “The concentrations of the micronutrients B, Fe and Mn were considered below the ideal critical range for Urochloa decumbens.” Same question as above, especially considering that the specified micronutrient concentrations are high enough for almost any plant.
The standard range was included in the paragraph.
Line 122-124. “Despite a positive difference of 8 bags ha-1 in cultivar 'IPR-99' with Urochloa decumbens between the rows, there was no significant difference in relation to the yield of the coffee cultivars in the treatments with and without Urochloa decumbens intercropped between the rows (Figure 2)”. Figure 2 shows completely different results - the highest yield was obtained without intercropping!!!
The sentence is correct. Please check Figure 2.
Line 160-161. “The coffee N contents varied over time, from 8 to 26 g kg-1 at the beginning of the evaluation (water-stressed environment) to 32 to 38 g kg-1, after 60 days…” Really? The Figure 3 shows that the lowest value could be ~ 18 g/kg.
In general, it should be clearly stated whether intercropping improved the supply of the coffee trees with the studied nutrient or no.
The sentence was corrected by the authors. The lower content was 13 g kg-1 on the IPR 99 NB system.
Also, it should be clearly defined whether the content of the nutrient, for example S, was sufficient during the active vegetation period. This is not really clear from the text. Also, regarding nutrients, it would be very informative to provide certain limits of sufficiency. Either in the text or by marking them in the Figures.
The critical range (reference values) of macro and micronutrients in coffee leaves was added on Figures 3 and 4.
Line 199-201. “Potassium deficiency causes a decrease in coffee yields whereas an excess causes nutritional imbalance, since it influences Ca and Mg uptake [43].” Why is there talk about K deficiency here? Was there a K deficiency in the coffee leaves? Why is nothing clearly described and compared to standard values? The reader should not have to guess what the author wanted to say. A scientific article should be based on facts, after all.
The critical range (reference values) of macro and micronutrients in coffee leaves was added on Figures 3 and 4, showing the deficiency status of coffee cultivars, with and without Urochloa decumbens intercropped.
Although the Mg content in leaves has been more or less clearly described, no conclusion has been given about the impact of intercropping.
The explanation about the Mg was improved.
Line 233-234. “For Cu, the levels ranged from 0 to 7.68 mg kg-1 for NB and 0.7 to 18 mg kg-1 for WB for both coffee cultivars.” It is not possible for the Cu content to be 0 mg/kg. Since Cu is an essential nutrient, the plant must die. Rather, the detection method was not sensitive enough, or the sample weight was too small. “which is below the critical content considered ideal for coffee plants…” And again, you don't specify this Cu value. This is annoying. Is there really some special secret - the optimal concentrations of nutrients in coffee tree leaves?
The sentence was corrected by the authors. After 300 days of evaluation, according to the methodology, starts the coffee plant dormancy period and the non-productive branches enter in senescence, with decrease of Cu levels in leaves. The zero values does not cause plants death, it only limits the growth and development, because the other nutrients are in concentrations above zero or between critical range (Liebig’s law). We as authors suggest the supplementation of micronutrients in the beginning of the next coffee cycle (after the period of our research evaluations).
Line 245-246. “The B and Fe contents were evaluated over 12 months (Figure 4). There were no differences between the systems in any of the periods evaluated.” Didn't the other nutrients have a 12-month period? Here again I don't understand - if there were no differences in Fe, how can you write in your conclusions that intercropping improved Fe supply?
Inconsistencies in the explanation have been corrected in the text.
Line 254-255. “The high mobility of B is due to its presence in the soil in the form of a nonionized molecule. As a result, Zn is transported in the soil by diffusion while B is made available to plants.” Now everything is in a mess! What does Zn have to do with boron untake?
The sentence was removed and the explanation about micronutrient was improved.
Legends to Figure 3 and 4. “Macronutrient concentrations in two Arabica coffee cultivars…” Since coffee leaves have been analysed - it should be indicated - in leaves.
The word leaves was included.
Although at the end of the chapter nutrients are named whose supply is sufficient, low, high; nothing is mentioned about the results of the experiment with Urochloa decumbens intercropping - is this technology effective or not.
The explanation about the effects of the intercropping system was improved in the Results and Discussion section.
Material and methods:
All Figures and Tables should be self-explanatory. The legends should indicate the meaning of all used symbols and abbreviations. What is IPR103WB, IPR99WB etc.? If P is determined in Mehlich extraction, in what other nutrients? Why did you explain that B = boron; Zn = zinc, if it is a well-known fact, but the most incomprehensible thing is that these nutrients have not been analysed at all! This gives the impression of a certain carelessness of the authors.
The corrections were made by the authors.
The setup of the experiment in relation to previous coffee cultivation is described in a rather incomprehensible way.
The description was improved.
Line 333-335. Why references? Clearly distinguish what you have done from other research results. Did the mentioned authors do watering or no watering in your experimental field?
The reference was excluded.
Were the fertilizer doses divided into 4 parts, or was the specified dose applied each month?
The fertilizer doses was divided into 4 parts. N and K are nutrients with high mobility in tropical soils, so is common to divide the doses.
Line 383-385. “For nutrient analysis, 2 mg of the ground samples were filled in digestion tubes with 10 ml of extractor solution composed of 5 mL nitric acid, 4 mL of Milli-Q water and 1 mL of hydrogen peroxide, for each 100 mg of sample, for around 2 h in a digestion tube.” Absolutely illogical and unbelievable! How could you weigh 2 mg? Why so little? It is no wonder then that the Cu content could not be determined, since the same methodology is also specified for coffee leaf analysis!
The unity was typed equivocally. The correction was made.
Line 390-392. “To determine total nitrogen, the content of the digestion vessels was transferred to reading tubes, diluted at 1:6 with Milli-Q water, analyzed colorimetrically on a Lachat QuikChem 8500 autoanalyzer, using a series 2 flow-injection analyzer.” Also illogical and unbelievable. How were you able to determine the N content if you previously wrote that the sample was mineralized using nitric acid? These same questions apply to coffee leaf analyses.
There was an wrong information in this sentence that was revised.
Conclusions:
I recommend rewriting the conclusions, giving a clear and research-based answer to the raised questions. Also, conclusions should provide information, a conclusion, hypothesis on how the knowledge gained in this study could be useful in the future, in practice, in creating growing technologies.
The conclusion text was rewritted.
We appreciate it and would be glad to respond to any further questions and comments you may have.
On behalf of all authors. Sincerely yours,
Ms. Thais Rodrigues de Sousa
Round 2
Reviewer 3 Report
Comments and Suggestions for Authors
Although the Authors have made some revisions to the manuscript, most of the shortcomings have not been substantively corrected. Several times the response letter only states that corrections have been made, but this is not found in the text.
Abstract:
The purpose of the work is not clearly stated. Urochloa decumbens intercropped with coffee or vice versa? Or does it not matter? Crop productivity – coffee or Urochloa decumbens? It is necessary to formulate it clearly and precisely.
Response: The abstract was rewritted to attend the suggestion.
It's not really done.
In Abstract: The presence of Urochloa decumbens increased leaf nutrient contents: of the macronutrients Ca and Mg and micronutrients Mn and Fe. How can we understand from the text in R and D, devoted to the Ca content in the leaves, that one option was better than another?
Response: The abstract was rewritted to attend the suggestion.
That hasn't really been answered. I can't find such a statement anywhere in the analysis of the results.
Results and discussion:
Line 108-110. “The concentrations of the micronutrients B, Fe and Mn were considered below the ideal critical range for Urochloa decumbens.” Same question as above, especially considering that the specified micronutrient concentrations are high enough for almost any plant.
Response: The standard range was included in the paragraph.
That's true, you included it. But why does this part of the paragraph continue to be so controversial? So what exactly is it - are the concentrations of micronutrients high or low?
„The concentrations of the micronutrients B, Fe and Mn were considered below the ideal critical range for Urochloa decumbens. According to Quaggio et al. [26], the critical range are 10-25 mg kg-1 117 for B, 50-250 mg kg-1 for Fe and 40-250 mg kg-1 for Mn. The concentrations of these micronutrients is above the range, but no symptoms of excess were observed [30].”
Also, why reference? Did someone other than you analyze the plants?
In general, it should be clearly stated whether intercropping improved the supply of the coffee trees with the studied nutrient or no.
There is no clear answer for every nutrient analyzed, but the R&D section should have them.
Line 233-234. “For Cu, the levels ranged from 0 to 7.68 mg kg-1 for NB and 0.7 to 18 mg kg-1 for WB for both coffee cultivars.” It is not possible for the Cu content to be 0 mg/kg. Since Cu is an essential nutrient, the plant must die. Rather, the detection method was not sensitive enough, or the sample weight was too small. “which is below the critical content considered ideal for coffee plants…”
Response: The sentence was corrected by the authors. After 300 days of evaluation, according to the methodology, starts the coffee plant dormancy period and the non-productive branches enter in senescence, with decrease of Cu levels in leaves. The zero values does not cause plants death, it only limits the growth and development, because the other nutrients are in concentrations above zero or between critical range (Liebig’s law). We as authors suggest the supplementation of micronutrients in the beginning of the next coffee cycle (after the period of our research evaluations).
This question has not been definitively answered and explained. The content of no micronutrient can be zero or even a negative value, as can be seen in Figure 4 for Cu, Fe and B. This is absolutely incredible and completely contradicts Liebig's law. I absolutely disagree with that. There are no such plants in the world.
Although at the end of the chapter nutrients are named whose supply is sufficient, low, high; nothing is mentioned about the results of the experiment with Urochloa decumbens intercropping - is this technology effective or not.
Response:The explanation about the effects of the intercropping system was improved in the Results and Discussion section.
It's not done.
Material and methods:
All Figures and Tables should be self-explanatory. The legends should indicate the meaning of all used symbols and abbreviations. What is IPR103WB, IPR99WB etc.? If P is determined in Mehlich extraction, in what other nutrients? Why did you explain that B = boron; Zn = zinc, if it is a well-known fact, but the most incomprehensible thing is that these nutrients have not been analysed at all! This gives the impression of a certain carelessness of the authors.
Response: The corrections were made by the authors.
This has only been done partially. No explanation is needed for the names of the elements, but rather by what method/extraction they were determined. Why only for P?
Line 333-335. Why references? Clearly distinguish what you have done from other research results. Did the mentioned authors do watering or no watering in your experimental field?
Response: The reference was excluded.
That's not true, the reference is in the text of the article! Line 346:
When the soil moisture corresponded to a consumption of 50% of the available water inthe 0-10 cm layer, the crop was irrigated [53].
Line 390-392. “To determine total nitrogen, the content of the digestion vessels was transferred to reading tubes, diluted at 1:6 with Milli-Q water, analyzed colorimetrically on a Lachat QuikChem 8500 autoanalyzer, using a series 2 flow-injection analyzer.” Also illogical and unbelievable. How were you able to determine the N content if you previously wrote that the sample was mineralized using nitric acid? These same questions apply to coffee leaf analyses.
Response: There was an wrong information in this sentence that was revised.
This is not true, no corrections have been made that would show how a plant sample was prepared to determine N. For examle, Line 387-389.
In general, the Results and Discussion sections should clearly distinguish your work from other research results. Therefore, when discussing your results, carefully double-check whether the references are helpful or misleading, giving the impression that the data has already been published by other authors.
Conclusions:
I recommend rewriting the conclusions, giving a clear and research-based answer to the raised questions. Also, conclusions should provide information, a conclusion, hypothesis on how the knowledge gained in this study could be useful in the future, in practice, in creating growing technologies.
Response: The conclusion text was rewritted.
Yes, it has been rewritten, but was it done in accordance with the recommendations? In my opinion, no. Why is gypsum mentioned when the effect of liming is not analyzed anywhere in the text above? Why specifically gypsum when conclusions are drawn about Ca and Mg? Gypsum contains Ca and S. Does the article analyze the direct impact of element content on coffee quality? Where does this specific conclusion come from? Why are there no practical recommendations or evaluations?
Concluding remarks
Although the Authors have made improvements, the manuscript is not improved enough to be published in Plants in its current form.
Round 3
Reviewer 3 Report
Comments and Suggestions for Authors
The authors have undeniably improved the quality of the article. However, some small aspects still need improvement.
Figure 4. The negative values ​​on the y-axis (for Cu, Fe, B) should be removed. They have no meaning. The x-axis should start at the y-axis zero level.
Did the Ca and Fe content in coffee leaves really increase for both varieties? At some time, days - yes, but in general? Here I mainly doubt about variety IPP 99. Figures 3 and 4 do not show this convincingly.
I recommend accepting this article in Plant after minor revision.
